



# Contribution of model parameter uncertainty to future hydrological projections

Qinghuan Zhang[1], Qiuhong Tang[1,2], John F. Knowles[3,4], Ben Livneh[5]

[1]Key Laboratory of Water Cycle and Related Land Surface Processes, Institute of Geographic Sciences and Natural Resources Research, Chinese Academy of Sciences, Beijing, China

[2]University of Chinese Academy of Sciences, Beijing, China

[3]USDA ARS Southwest Watershed Research Center, Tucson, AZ, USA

[4]Institute of Arctic and Alpine Research, University of Colorado Boulder, Boulder, CO, USA

[5]Department of Civil, Environmental, and Architectural Engineering, University of Colorado Boulder, Boulder, CO, USA

Correspondence to: Qinghuan Zhang (zhangqh@igsnrr.ac.cn)

**Abstract.** Hydrologic models have been applied to predict land surface water and energy budgets in mountainous watersheds that are characterized by complex geological features and climatic variability. A common practice is to calibrate the models and achieve the best performing parameter set according to historical observations, and then the calibrated model was used to do future projections. One drawback is

that the influence of parameter uncertainty on model projections is not well discussed. In this study, we applied multiple objective functions to choose a group of best performing parameter sets to the Boulder Creek Watershed, USA to investigate how parameter uncertainties can propagate to future projections. We used 16 parameter sets that have similar performance in simulating streamflow amount and regime historically, and applied the same parameter sets to predict hydrologic variables including streamflow,

evapotranspiration, and soil moisture in two future phases (Phase 1 is 2040-2069 and Phase 2 is 2070-2099). The results show that variability due to parameter uncertainty was up to 10 % annually and 26 % monthly under future climate change scenarios, and the uncertainties are especially prominent during May to September. The different parameter sets can result to annual streamflow changes in opposite directions. The results indicate that a single parameter set may yield biased representation of hydrologic variability.

It is necessary to consider multiple optimal parameter sets in applying hydrologic models for hydrological projections and water resources decision making.



## 1. Introduction

Streamflow variability in the mountains is influenced by factors such as the frequency, magnitude, and type of precipitation, land cover characteristics, snow accumulation and ablation, and the timing and rate of snowmelt (Harding et al., 2012; Barnhart et al., 2016; Musselman et al., 2017; Zhang et al., 2018). Future climate change is predicted to impact all of these factors with implications for regional water availability to municipal, industrial, mining, irrigation, hydropower generation (Wagener et al., 2010; Bates et al., 2008; Barnett et al., 2005; Liu et al., 2016; Yin et al., 2017). Further, hydrological impacts due to warming are accentuated at high elevations because water yields are larger and the processes that dictate snow accumulation and melt are sensitive to changes in air temperature (Gao et al., 2010; Bales et al., 2006). The Intergovernmental Panel on Climate Change (IPCC) has specifically stated that current water management systems cannot cope with the impacts of climate change, and that significant drought and flood damage is expected to occur (Bates et al., 2008). However, given accurate information, land managers can develop targeted watershed management plans to adapt to and potentially mitigate the effects of climate change on streamflow and ecosystem function (Watts et al., 2016; Poff et al., 1997). Robust characterization of streamflow conditions under future climate scenarios is essential for assessing the sustainability of the long-term supply of water to the natural and managed system downstream (Pradhanang et al., 2013).

Hydrological models can improve our understanding of land surface water and energy budgets in mountainous areas by simulating the future hydrological cycle as a function of climate forcing data from Global Climate Models (GCMs) (Dessu and Melesse, 2013). Downscaled climate data from General Circulation Models (GCMs) are frequently used within a hydrologic model to predict how changes in climate affect the water balance using a variety of multi-scale approaches and analyses (Wilby et al., 2009; Liu et al., 2017). Previous studies have shown that the magnitude of projected climate perturbation represents a major source of uncertainty associated with hydrologic variables obtained from hydrologic models (Schewe et al., 2014; Mendoza et al., 2015; Raje and Krishnan, 2012; Guo et al., 2018). In addition, different GCMs, emission scenarios, and the time period over which the climate perturbation was obtained, can all affect the magnitude and direction of hydrologic projections (Addor et al., 2014). Previous studies have specifically considered hydrological models in the context of a "cascade of uncertainty" to compare its contribution to total uncertainty with the other elements of the model chain, which provides valuable insights into the development of hydrological modeling in changing conditions (Addor et al., 2014). Further, hydrologic response of one particular type of catchments to climate change may be representative of similar types (Köplin et al., 2012). The uncertainties in hydrologic modeling from parameter sets are not well discussed. Mendoza et al. (2016) mentioned that parameter sets with





different locations in the parameter space can have different signals when projecting future streamflow conditions. The traditional measure of performance is a single objective function, which may not be sufficient to discriminate between competing models (Wagener, 2003). The use of multiple objective functions to calibrate model parameters can help to partition total runoff between direct runoff and
baseflow (Liu et al., 2018), and to overcome the problem of lacking detail in conceptual models (Gupta et al., 2008). As a result, this work is focused on the influence of parameter uncertainty on modeled hydrological outputs in a snowmelt-dominated watershed, toward an improved understanding of how uncertainty affects model projections under future climate change scenarios.

This work specifically applied the distributed, physically-based Variable Infiltration Capacity
(VIC) hydrologic model to a regionally important (Boulder Creek) watershed. The specific objectives of the study were to: (1) quantify the uncertainty resultant from model parameters to projections of hydrologic flux and state variables, and to (2) assess the time-dependency of this uncertainty within the context of hydrological change within the Boulder Creek Watershed. As human activity is increasing and human-induced climate warming is occurring, appropriate management strategies must be based on
reliable predictions of freshwater occurrence, circulation, distribution, and quality under a perturbed climate such as that provided by this work.

## 2. Study area

The Boulder Creek Watershed at Orodell is located in the southern Rocky Mountains, USA and has an area of 264 km². The elevation ranges from 1779 m to 4117 m, with a mean elevation of 3139 m.
The mean annual precipitation is 840 mm. Orodell is located downstream of the confluence of Middle and North Boulder Creek at an elevation of 1779 m, with coordinates of 40.006° N and 105.33° W (USGS stream gauge 6727000). In general, low-flow at the watershed occurs from October to March, while high-flow occurs from May to July and peaks in June, depending on snowpack depth and air temperature (Murphy et al., 2003). Soil cover is generally thin and thus streamflow is relatively independent of soil
moisture conditions (Rauscher et al., 2008). However, soil water is maintained through the sustained input of snowmelt water during the spring, which promotes significant runoff during wet periods (Hamlet and Lettenmaier, 2007). The geographical location of the watershed is shown in Figure 1.



## 3. Data and methods

### 3.1 The Variable Infiltration Capacity (VIC) Model

#### 3.1.1 Model mechanism

The VIC model is a physically-based, distributed, macro-scale hydrologic model that is capable of simulating the water balance from distributed grid cells through calculations of runoff, baseflow, evapotranspiration (ET), snow water equivalent, and other variables (Liang et al., 1994). It has been applied to assess the impact of climate change in many river basins in the western United States (Harding et al., 2012; Barnhart et al., 2016). The grid cells are modeled as flat surfaces and sub-grid heterogeneity is handled via statistical distributions. No horizontal routing of surface overland flow, subsurface flow, or channel flow is performed (Mendoza et al., 2015), although streamflow routing can be post-processed through a routing model (Lohmann et al., 1996). Surface runoff at each time step is defined as precipitation that exceeds the storage capacity of the soil at the previous time step. Baseflow is defined as a function of the soil moisture in the third soil layer (Arno formulation) from the surface, which is linear below a threshold and non-linear above that threshold (Gao et al., 2010). Total runoff is the sum of surface runoff and baseflow.

#### 3.1.2 Historical data input and streamflow

The VIC model simulations were driven by inputs of daily gridded precipitation, maximum and minimum air temperature, and wind speed, and were calibrated against streamflow observations between water years (1 Oct – 30 Sep) 1981 to 1990. In total, eight parameters were calibrated at 1/8° spatial resolution (Table 1). The parameter ranges were selected following Demaria et al. (2007) for the contiguous United States and former modeling experiments at this watershed. It is assumed that these parameters are constant in time and representative of inherent properties of this watershed. The climate forcing data was obtained from Maurer et al. (2002) at 1/8° spatial resolution.

Naturalized daily streamflow data from 1906 to 2011 were combined from the United States Geological Survey at station number 6727000 and the Colorado Division Support System at gage BOCOROCO. Both of the streamflow gauges are located at Orodell. Naturalized flow represents flow that would have occurred at the stream gauge had historical depletions and reservoir regulation not been present (Harding et al., 2012). Simulated streamflow from 1981 to 2010 was applied as the baseline level for subsequent analyses. Since both model parameters and climate forcing inputs could influence hydrologic simulations (Ren et al., 2016), a time series including wet and dry climatic conditions is necessary when calibrating models.





### 3.1.3 Land cover and soil parameters

Land cover types were obtained from the United States Geological Survey National Gap Analysis Program and were reclassified into 11 classes according to the 1-km Global Land Cover Classification at the University of Maryland (Hansen et al., 2000). Leaf area index (LAI), in particular, has been shown to influence interception, evaporation, and runoff generation (Dietz et al., 2006; Chen et al., 2005). In our study, the annual LAI was held constant through time but was allowed to vary on a monthly basis. For each vegetation type, monthly LAIs were applied from the VIC vegetation library file.

Soil parameters at 1/8° spatial resolution including saturated hydrologic conductivity, mineral type, fractional soil moisture at the critical point and at the wilting point, and soil bulk density were compiled from the STATSGO database produced by the United States Department of Agriculture (Soil Survey Staff, 2015). In STATSGO, Map Unit Delineation consists of one or more closed polygons that are generally geographic mixtures of groups of soils or soils and non-soil areas. Each map unit can have as many as 21 components. A component is a phase of a soil series (which is the lowest category of the national soil classification system) representing the most homogenous classes in the taxonomy system. Each component can have multiple soil layers and the size of each component is provided by areal percentage of the map unit (Huang, 2005). In this study, the dominant soil class in each grid cell was applied to calculate soil parameters for that grid cell.

### 3.1.4 Model calibration and performance evaluation

Model parameters were calibrated by comparing observed streamflow with simulated streamflow from 1981 to 1990; the two years prior to 1981 was used for model spin-up (Shi et al., 2008). The period 1991 to 2010 was used for model validation. We calibrated the model by comparing simulated and observed streamflow data using the Borg Multi-Objective Evolutionary Algorithm (MOEA), which is an iterative search algorithm for many-objective optimization problems (Hadka and Reed, 2013). Sampling with MOEA is performed via the random seed method, where the parameters are selected randomly from a parameter range by optimizing the user-specified multiple objective functions (Hadka and Reed, 2013; Hadka and Reed, 2015). In the calibration process, the number of total iterations was set to 10 000. Model performance was evaluated using the Nash-Sutcliffe efficiency (NSE), correlation coefficient ($R^2$), root mean square error (RMSE), percent bias (PBIAS), and the ratio of standard deviations of simulations to observations (RSD) (Table 2).

### 3.2 Global climate models and future scenarios

The selection of a particular climate model simulation affects water yield predictions under climate change (Stone et al., 2003) and may give biased results. Hence, the use of multiple climate



models constrains the possible range of hydrologic change. The Coupled Model Intercomparison Project Phase 5 (CMIP5) dataset compares multiple global climate models under four future emission scenarios. The four emission scenarios, also called representative concentration pathways (RCPs), are based on concentration pathways that approximate various level of radiative forcing (W m$^{-2}$) at the end of the 21$^{st}$

century (Meinshausen et al., 2011). Specifically, the greenhouse gas concentration trajectories RCP 2.6 (Vuuren et al., 2011), 4.5 (Thomson et al, 2011), 6 and 8.5 (Riahi et al, 2011) are based on radiative forcing values of 2.6, 4.5, 6 and 8.5 W m$^{-2}$, respectively, in the year 2100. In particular, the RCP 8.5 scenario assumes high population growth and high-energy demand without implementation of policy to address climate change. In this study, eighteen CMIP5 GCMs were applied in RCP 8.5 (Table 3) for the

purpose of simulating hydrologic responses to future climate projections using the VIC model. Hydrologic predictions (including streamflow, evapotranspiration as ET, and soil moisture) under future climate change were compared with baseline simulations in historical period. The ensemble mean of the eighteen climate models was also applied as an additional climate dataset to represent the average condition from the individual models. The climate datasets were statistically downscaled using the Bias

Corrected Spatial Downscaling (BCSD) method and were obtained from the United States Bureau of Reclamation (Reclamation, 2013).

### 3.3 The delta-change method

The delta-change method computes the difference between the current and future simulations of air temperature and precipitation and adds these changes to the observed time series (Wood et al., 1997;

Hay et al., 2000; Fowler et al., 2007). This approach assumes that GCMs more accurately simulate relative change than absolute values (i.e., there is a constant bias through time). The delta-change method uses the initial input forcing datasets and the GCMs change signal, which equals the observations under current conditions and outperforms the other bias correction methods (Teutschbein and Seibert, 2012). Hay et al. (2000) compared several downscaling methods and determined that the delta-change method

provided more conservative estimates of changes in future runoff. The current study applied the delta-change method to existing downscaled GCM data to obtain a new dataset for each GCM model. Since the variability in internal parameterizations from GCMs results in significant uncertainty, the use of model ensembles can provide a realistic assessment of climate change (Fowler et al., 2007). Two future time periods in RCP 8.5 were considered: Phase 1 (2040-2069) and Phase 2 (2070-2099), which are

representative of the middle and the end of the 21$^{st}$ century, respectively.

The following workflow describes the steps that were performed to generate the precipitation and air temperature datasets that were acted as input forcing data for VIC: (1) Climate data from eighteen GCMs using the BCSD downscaling method were obtained from the United States Bureau of





Reclamation (Reclamation, 2013); (2) For each month in the historical time period 1981-2010, monthly average values for precipitation and maximum and minimum air temperature from each CMIP5 climate model were calculated and averaged over the study area; (3) Monthly average values of precipitation, maximum and minimum air temperature of the initial climate data (Maurer et al., 2002) were calculated during the historical time period (1981-2010); (4) The delta change was calculated as the difference in air temperature (℃) or the ratio of precipitation (%) between the initial climate data and each of the eighteen CMIP5 climate models; (5) The delta values from each model were applied to the initial daily forcing data. Calculations of the new future climate data are shown in Equations (1) (minimum and maximum air temperature) and (2) (precipitation):

$$T_{fut,d-m} = T_{fut\_GCM,d-m} + DeltaT_m \quad (1)$$

$$P_{fut,d-m} = P_{fut\_GCM,d-m} \times RatioP_m \quad (2)$$

where $T_{fut,d-m}$ and $P_{fut,d-m}$ represent the projected temperature (℃) and precipitation (mm) for day d in month m (m = 1, …, 12), $T_{fut\_GCM,d-m}$ and $P_{fut\_GCM,d-m}$ represent the CMIP5 temperature (℃) and precipitation (mm) for day d in month m in the future, $DeltaT_m$ represents the air temperature delta change (℃) for month m, and $RatioP_m$ represents the precipitation ratio (%) of the initial data to CMIP5 precipitation for month m. In this way, we generated eighteen new future climate datasets corresponding to the BCSD downscaled climate data.

### 3.4 Percent change and uncertainty analysis

Percent changes in future precipitation, streamflow, and soil moisture were calculated as

$$Delta\ X\ (\%) = \frac{(X_{future} - X_{historical})}{X_{historical}} \times 100\% \quad (3)$$

where $X_{future}$ represents precipitation, streamflow, or soil moisture for a future time period and $X_{historical}$ represents precipitation, streamflow, or soil moisture in the historical time period. For each of these variables, the median value was calculated for each parameter set and for all climate data. The range of the median values from all parameter sets was then divided by the total range of the projected variable using all parameter sets and climate data (Table 4).

## 4. Results

### 4.1 Model calibration results

The VIC model was calibrated against the observed streamflow at the Orodell stream gauge. The 16 best performing parameter sets were chosen randomly using the Borg MOEA framework. Figure 2 shows the parameter sets with daily NSEs. All parameters except soil depth 1 were sensitive to streamflow simulations as evidenced by decreasing scatter with increasing NSE. Figure 3 shows the




relationship between NSE during the calibration period and other calibration metrics. The RMSE was inversely correlated with NSE, indicating autocorrelation and thus redundancy of the RMSE metric whereby the minimization of RMSE is equivalent to the maximization of NSE (Mendoza et al., 2016). The $R^2$ values were directly proportional to NSE, whereas the PBIAS and RSD values demonstrated a

non-linear relationship to NSE. The optimal parameters of the RSDs and NSEs form a Pareto Front where the best performing parameters along this front have no preference over one another (Hadka and Reed, 2013). As expected, the NSEs during the validation time period were lower than during the calibration time period, but the NSEs during the validation and calibration time periods were correlated, suggesting that the performance of a given parameter set may be transferrable across time periods. Among the

optimal parameter sets, the NSE values varied from 0.745 to 0.779. The ratio of standard deviations (RSD) ranged from 0.689 to 0.994, percent bias (PBIAS) ranged from 0.0008 to 6.594, $R^2$ ranged from 0.76 to 0.79, and RMSE ranged from 0.43 to 0.46.

      The selected parameters were subsequently used to simulate streamflow (Figure 4). During the validation period (1991 to 2010), daily NSEs ranged from 0.549 to 0.687 and averaged 0.617. Although

there were discrepancies between flow volumes at peak flow, the timing of observed streamflow was well predicted by the model (Figure 4a). Figure 4b presents the daily simulated and observed streamflow on a log scale to emphasize the low flow period when different parameter sets produced a large uncertainty range.

### 4.2 Precipitation and air temperature changes under perturbed climate

The delta change method was applied to each GCM model using BCSD downscaled data. Daily precipitation and maximum and minimum air temperature between 2040-2069 (Phase 1) and 2070-2099 (Phase 2) were compared with the historical period 1981-2010 (Figure 5). Different climate models show various trends and magnitudes in changes in average annual precipitation, but similar trends in maximum and minimum temperatures. The majority of models predicted higher precipitation in Phase 2 relative to

Phase 1. Changes in annual precipitation varied from -5.4 % to +29.2 % and averaged +8.7 % in Phase 1 compared to the historical period. In phase 2, changes in average annual precipitation varied from -2.7 % to +31.0 % and averaged +12.7 % in Phase 2 compared to the historical period. In contrast, all models showed increasing trends in maximum and minimum air temperatures. Changes in daily maximum temperature varied from +1.7 ℃ to +3.9 ℃ with an average of +2.9 ℃ in Phase 1, and from +3.1 ℃ to

+6.8 ℃ with an average of +5.0 ℃ in Phase 2. Changes in daily minimum temperature varied from +1.7 ℃ to +3.9 ℃ with an average of +2.9 ℃ in Phase 1, and from +3.3 ℃ to +6.3 ℃ with an average of +4.8 ℃ in Phase 2.





### 4.3 The water balance

Changes in average annual streamflow, ET, and soil moisture in Phases 1 and 2 with respect to the historical period were calculated. Figure 6 presents the median values of percent changes in average annual streamflow, ET, and soil moisture using each parameter set and all climate data input. The dashed lines represent the median values using all parameter sets and climate data. Generally, annual streamflow will increase by 0.1 % but decrease by 0.9 % in Phases 1 and 2, respectively (Figure 6a). The different parameter sets show that streamflow will change by -1.1 % to 1.9 % in Phase 1 and change by -4.6 % to 3.2 % in Phase 2. Annual ET will increase by 9.6 % and 18.2 % in Phases 1 and 2 (Figure 6b). The different parameter sets show that ET will increase by 9 % to 9.7 % in Phase 1 and increase by 17 % to 18.9 % in Phase 2. Soil moisture will decrease by 2.9 % and 5.1 % in Phases 1 and 2, respectively (Figure 6c). The different parameter sets show that soil moisture will decrease by 2.3 % to 3.7 % in Phase 1 and decrease by 2.5 % to 5.5 % in Phase 2.

The percent ranges of median changes using each parameter set were 3.8 % and 10.2 % for annual streamflow, 3.3 % and 6.2 % for ET, and 5 % and 10.8 % for soil moisture with respect to the total ranges using all parameter sets and climate data for Phases 1 and 2 respectively (Table 4).

### 4.4 Monthly changes in flux and state variables

The average monthly values of streamflow, ET, and soil moisture during the historical and future periods are presented in Figure 7. Using the ensemble mean climate data, uncertainty in streamflow and ET predictions was higher between May and August than in the other months, whereas the soil moisture uncertainty was relatively consistent throughout the year.

Boxplots of percent changes in monthly values and boxplots using median values of percent changes from historical to Phases 1 and 2 were shown in Figure 8. In general, streamflow showed an increasing trend between February and June, and a decreasing trend from July to November in Phase 1. In Phase 2, streamflow showed an increasing trend between February and May, and a decreasing trend from June to December. The magnitudes of increases were the highest in May in Phase 1 and in April in Phase 2, respectively (Figure 8a). ET showed an increasing trend in all months, while the increasing magnitudes were lower in June to September than in the other months (Figure 8b). Soil moisture showed an increasing trend in February to May, and a decreasing trend in other months in the two phases (Figure 8c). The magnitude and variability of changes were generally higher in Phase 2 than in Phase 1 for streamflow, ET, and soil moisture.

Monthly changes in the streamflow uncertainty were higher in February to April, June, and August than in the other months in Phase 1, while the uncertainties were higher in March to June than in the other months in Phase 2 (Figure 8d and 8g). The uncertainty associated with monthly changes in ET





was higher in the period July to September than in the other months (Figure 8e and 8h). Monthly changes in soil moisture uncertainty were higher in January to May than in the other months in Phases 1 and 2, while the uncertainties also increased in July to September in Phase 2 (Figure 8f and 8i).

## 5. Discussion

This study applied the VIC model to the Boulder Creek Watershed to analyze the impacts of predicted climate change on hydrologic flux and state variables. The selected model parameters were calibrated by comparing simulated streamflow against observed streamflow using the multiple objective optimization mechanisms. Climate input forcing data in the future applied BCSD downscaled data from eighteen GCM models. The results from all climate model inputs and parameter sets show that in general, annual streamflow will increase by 0.1 % but will decrease by 0.9 %, annual ET will increase by 9.6 % and 18.2 %, and soil moisture will decrease by 2.9 % and 5.1 % in Phases 1 and 2, respectively. Different parameter sets resulted in up to 10 % in annual and 26 % in monthly projection uncertainties.

### 5.1 Multiple objective functions and climate data processing

All of the chosen parameter sets had NSE values of greater than 0.7, which is considered good model performance (Moriasi et al., 2007). The choice of objective functions may be necessary to consider non-autocorrelation. For example, RMSE and NSE are closely correlated, and therefore may be redundant. The choice of many parameters provides general information about parameter sensitivity, which constrains uncertainty in streamflow and facilitates the transfer of model parameters from data-rich to data-sparse areas (e.g. Huang 2005). In this case, the calibrated parameters all converged to a small range at the best model performance except soil depth 1. The parameter b and soil depth 1 may be important for high-flows, while the other calibrated parameters may be important for low-flows or baseflow. Over the long-term, the simulated monthly trends in streamflow represented observed values during the historical period (Figure 7).

Although the delta change method removed large variations in climate data from the GCMs, it also assumes that the magnitude of changes in precipitation and temperature remain stationary. Hewitson and Crane (2006) asserted that the degree of non-stationarity in projected climate change is relatively small, and that circulation dynamics in particular may be more robust to non-stationarities. Gutmann et al. (2014) used climate data to compare several statistical downscaling methods and found that precipitation from the BCSD method is biased low for individual months, but unbiased on an annual scale. In this study, BCSD downscaling was used to show that average annual precipitation and maximum and



minimum air temperature will increase in both Phases 1 and 2, but that the increase in magnitude will be higher in Phase 2 than in Phase 1 (Figure 5).

### 5.2 Uncertainty in projected annual hydrologic fluxes

Although average annual precipitation is projected to increase by 8.7 % and 12.7 % in Phases 1 and 2, respectively, average annual streamflow shows no significant change in either Phase 1 (0.1 %) or Phase 2 (-0.9 %) compared to the historical period. The results showed that average annual ET increases by 9.6 % (Phase 1) and 18.2 % (Phase 2), while soil moisture decreases by 2.9 % (Phase 1) and 5.1 % (Phase 2). This indicates that the impact of increased air temperature on water resources availability may supersede that of precipitation changes in the future, as increasing temperature translated to increased ET. For each parameter set, simulation uncertainties were higher in Phase 2 than in Phase 1 (Figure 6). The different directions and magnitudes of changes in streamflow and soil moisture using various parameter sets indicate that using only one parameter set may give a biased result. Though different parameter sets resulted in a more consistent average annual ET trend, i.e., an increase in Phases 1 and 2, changes in average annual streamflow were uncertain (Figure 6). Overall, these results show that using median values from the best performing parameter sets can result in up to a 10 % uncertainty range in the corresponding model simulations (Table 4). Though annual streamflow demonstrated no significant change in Phase 1, the different parameter sets produced between a 1.9 % increase and a 1.1 % decrease. In Phase 2, annual streamflow is projected to decrease by 4 % where the different parameter sets ranged from a 3.2 % increase to a 4.5 % decrease. Consequently, the uncertainties resulted from parameter sets can have significant role in water supply and prediction.

### 5.3 Monthly projections

In snowmelt dominated watersheds, reservoirs are replenished in spring and water is released for agricultural, industrial, and urban water use during the summer (Ficklin et al., 2013). Using different parameter sets, the monthly streamflow and ET projections generally diverged during the summer, whereas soil moisture showed a relatively consistent but large range in all months. Parameter uncertainty causes a large variation for streamflow in June historically, and an equally large variation in June in the future periods considered by this work. Parameter uncertainty causes a small streamflow uncertainty in April historically, and an equally small variation in future periods. The uncertainties in monthly changes increase in March to May than the uncertainties in the other months.

The ET values were higher in Phase 2 than in Phase 1 or the historical period due to increased air temperature, which could also decrease the snow to rain ratio during the spring and therefore contribute to the higher spring streamflow. In summer, the combination of increased air temperature and decreased





precipitation amount could result in higher ET but lower streamflow in Phase 2 relative to Phase 1. Changes in soil moisture increased between March to May and decreased thereafter.

Ficklin et al. (2013) used three different modeling approaches to project streamflow under the A2 emission scenario in the UCRB and suggested that spring and summer streamflow is likely to decline by the end of the 21$^{st}$ century. The current results in the Boulder Creek Watershed show that on average, streamflow will increase from March to May but then decrease from June to August (Figure 8a). Further, the median monthly changes using different parameter sets result in a range of projected variables that allows for characterization of the uncertainty associated with predictions of both hydrologic flux and state variables. In particular, the uncertainty for changes in streamflow and soil moisture was higher in spring than in the other months, whereas the uncertainty for changes in ET was higher during the summer compared to the other months (Figure 8). In the context of Köplin et al. (2012) that used a cluster analysis to reduce 186 catchments to 7 response types in Switzerland, this study from a snowmelt dominated watershed may provide additional insights into other similar snowmelt dominated basins where hydrologic predictions may be unconstrained without uncertainty ranges resultant from the use of multiple parameter sets. These results demonstrate the need to apply multiple optimal parameter sets in order to make meaningful projections of water resource availability into the future.

## 6. Conclusion

Models are useful tools with which to evaluate the potential impacts of climate change on hydrologic variables. Though climate data inputs are critical to deciding model results, parameter uncertainty can also result in projection uncertainties. This study applied multi-objective optimization functions to calibrate the VIC model in the Boulder Creek Watershed at 1/8° spatial resolution. The optimal parameter sets were subsequently applied to simulate various water balance components in two future time periods. Using the ensemble mean climate data, average annual precipitation is predicted to increase by 8.7 % and 12.7 % from 2040-2069 (Phase 1) and from 2070-2099 (Phase 2) compared to the historical period, respectively. The median values from all models show that annual streamflow has no significant change, ET is projected to increase by 9.6 % and 18.2 %, while soil moisture is projected to decrease by 2.9 % and 5.1 % in Phases 1 and 2, respectively. The corresponding uncertainty analysis using different parameter sets further provides a range of expected flux and state characteristics during both Phase 1 and Phase 2. Parameter uncertainty has an important role in deciding changes in future hydrologic variables annually (up to 10.8 %) and seasonally (up to 26 %), and especially for streamflow and soil moisture. These results constrain the uncertainty resultant from model parameter sets with implications for water resources allocation and supply.




*Data availability.* The climate forcing data in historical period are freely available from Maurer et al. (2002) (http://www.engr.scu.edu/~emaurer/gridded_obs/index_gridded_obs.html) at 1/8° spatial resolution. The BCSD climate data are available from United States Bureau of Reclamation (http://gdo-dcp.ucllnl.org/downscaled_cmip_projections/#Projections:%20Complete%20Archives; Reclamation, 2013).

*Author contributions.* QZ, QT and BL designed the study. QZ performed the analysis and wrote the manuscript. JK, QT, and BL commented on the manuscript.

*Competing interests.* The authors declare that they have no conflict of interest.

*Acknowledgements.* This research is partly supported by the National Natural Science Foundation of China (41730645, 41790424 and 41425002), the Strategic Priority Research Program of Chinese Academy of Sciences (XDA20060402) and International Partnership Program of Chinese Academy of Sciences (131A11KYSB20170113). We acknowledge computing time on the University of Colorado CSDMS High-Performance Computing Cluster. We also acknowledge the World Climate Research Programme's Working Group on Coupled Modeling, which is responsible for CMIP, and we thank the climate modeling groups (listed in Table 3 of this paper) for producing and making available their model outputs. The United States Department of Energy's Program for Climate Model Diagnosis and Intercomparison provided coordinating support for CMIP and led development of software infrastructure in partnership with the Global Organization for Earth System Science Portals. The authors also thank Pablo Mendoza for his former suggestions.

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





7    Table captions

8    **Table 1. Summary of VIC parameters selected for calibration at Boulder Creek**

| Parameter | Description | Units | Calibration range | |
| :---: | :---: | :---: | :---: | :---: |
| | | | Min | Max |
| b | Variable infiltration curve parameter | Fraction | 0.001 | 0.8 |
| Ds | Fraction of Dsmax where nonlinear baseflow begins | Fraction | 0.001 | 1 |
| Ds_max | Maximum velocity of baseflow | mm/day | 1 | 20 |
| Ws | Fraction of maximum soil moisture where nonlinear baseflow occurs | Fraction | 0.001 | 1 |
| c | Exponent used in baseflow curve | - | 1 | 4 |
| Depth 1 | Thickness of soil layer 1 | m | 0.01 | 0.5 |
| Depth 2 | Thickness of soil layer 2 | m | 0.1 | 2 |
| Depth 3 | Thickness of soil layer 3 | m | 0.1 | 2.9 |




9    **Table 2. Evaluating functions of streamflow simulations at Boulder Creek**

| Formula | Name of indicator | Perfect simulation value |
|---|---|---|
| $NSE = 1 - \dfrac{\sum(Q_{ob,i} - Q_{si,i})^2}{\sum(Q_{ob,i} - \overline{Q}_{ob})^2}$ | Nash-Sutcliffe efficiency | 1 |
| $PBIAS = \left\| \dfrac{\sum(Q_{ob,i} - Q_{si,i})}{\sum Q_{ob,i}} \right\| \times 100$ | Percent bias | 0 |
| $RMSE = \sqrt{\dfrac{\sum_{i=1}^{n}(Q_{ob} - Q_{si})^2}{n}}$ | Root mean square error | 0 |
| $R^2 = \dfrac{\sum((Q_{ob,i} - \overline{Q}_{ob})(Q_{si,i} - \overline{Q}_{si}))}{\sqrt{\sum(Q_{ob,i} - \overline{Q}_{ob})^2 \sum(Q_{si,i} - \overline{Q}_{si})^2}}$ | Correlation coefficient | 1 |
| $RSD = \dfrac{\sqrt{\dfrac{1}{n}\sum(Q_{si,i} - \overline{Q}_{si})^2}}{\sqrt{\dfrac{1}{n}\sum(Q_{ob,i} - \overline{Q}_{ob})^2}}$ | Ratio of standard deviations | 1 |

10    Note: $Q_{ob,i}$-observed streamflow at time step i (mm/day); $Q_{si,i}$-simulated streamflow at time step i (mm/day); $\overline{Q}_{ob}$-

11    mean observed streamflow (mm/day); $\overline{Q}_{si}$-mean simulated streamflow (mm/day)



12    **Table 3. General climate models used in this study**

| Model ID | Name | Institution |
|---|---|---|
| CCSM4 | CCSM4 | National Center for Atmospheric Research, USA |
| CNRM | CNRM | National Centre for Meteorological Research, France |
| CSIRO | CSIRO | Commonwealth Scientific and Industrial Research Organization, Queensland Climate Change Centre of Excellence, Australia |
| CanESM2 | CESM2 | Canadian Centre for Climate Modeling and Analysis, Canada |
| CSM1 | CSM1 | Beijing Climate Center, China Meteorological Administration |
| CSM1m | CSM1m | Beijing Climate Center, China Meteorological Administration |
| GFDLG | GFDLG | NOAA Geophysical Fluid Dynamics Laboratory, USA |
| GFDLM | GFDLM | NOAA Geophysical Fluid Dynamics Laboratory, USA |
| HadGEM2C | HGEMC | Met Office Hadley Center, UK |
| HadGEM2E | HGEME | Met Office Hadley Center, UK |
| MIROC-ESM | MIROC | Japan Agency for Marine-Earth Science and Technology, Atmosphere and Ocean Research Institute (The University of Tokyo), and National Institute for Environmental Studies, Japan |
| MIROC5 | MIROC5 | Atmosphere and Ocean Research Institute (The University of Tokyo), National Institute for Environmental Studies, and Japan Agency for Marine-Earth Science and Technology, Japan |
| MIROC-CHEM | MIROCC | Japan Agency for Marine-Earth Science and Technology, Atmosphere and Ocean Research Institute (The University of Tokyo), and National Institute for Environmental Studies, Japan |
| INM-CM4 | INMCM | Institute for Numerical Mathematics, Russia |
| IPSL-CM5A-MR | IPSLA | Institut Pierre-Simon Laplace, France |
| IPSL-CM5B-LR | IPSLB | Institut Pierre-Simon Laplace, France |
| MRI-CGCM3 | MRI | Meteorological Research Institute, Japan |
| NorEMS1-M | NorESM | Norwegian Climate Center, Norway |





13 **Table 4. Percent range of the median changes using all parameter sets compared to the total range for**
14 **average annual and monthly streamflow (Q), evapotranspiration (ET), and soil moisture.**

| Variable | Time period | Annual | Jan | Feb | Mar | Apr | May | Jun | Jul | Aug | Sept | Oct | Nov | Dec |
|---|---|---|---|---|---|---|---|---|---|---|---|---|---|---|
| Δ Q | 2040-2069 | 3.8 | 5.1 | 10.3 | 6.1 | 3.9 | 2.8 | 10.0 | 10.8 | 19.2 | 14.7 | 10.6 | 5.2 | 5.2 |
| | 2070-2099 | 10.2 | 3.0 | 8.5 | 10.1 | 8.8 | 8.9 | 19.1 | 19.7 | 26.1 | 19.1 | 14.4 | 9.5 | 7.3 |
| Δ ET | 2040-2069 | 3.3 | 2.1 | 1.4 | 2.4 | 1.2 | 2.2 | 5.2 | 10.8 | 10.3 | 13.7 | 3.4 | 1.7 | 2.4 |
| | 2070-2099 | 6.2 | 1.7 | 2.3 | 2.8 | 4.1 | 6.4 | 20.5 | 12.1 | 18.9 | 19.3 | 11.5 | 7.5 | 2.3 |
| Δ Soil moisture | 2040-2069 | 5.0 | 14.3 | 15.6 | 13.4 | 8.4 | 12.0 | 3.0 | 5.7 | 6.7 | 9.7 | 9.9 | 4.9 | 11.1 |
| | 2070-2099 | 10.8 | 12.8 | 23.3 | 21.1 | 14.2 | 23.8 | 3.8 | 19.2 | 21.4 | 18.2 | 8.3 | 6.7 | 7.4 |





15    Figure captions

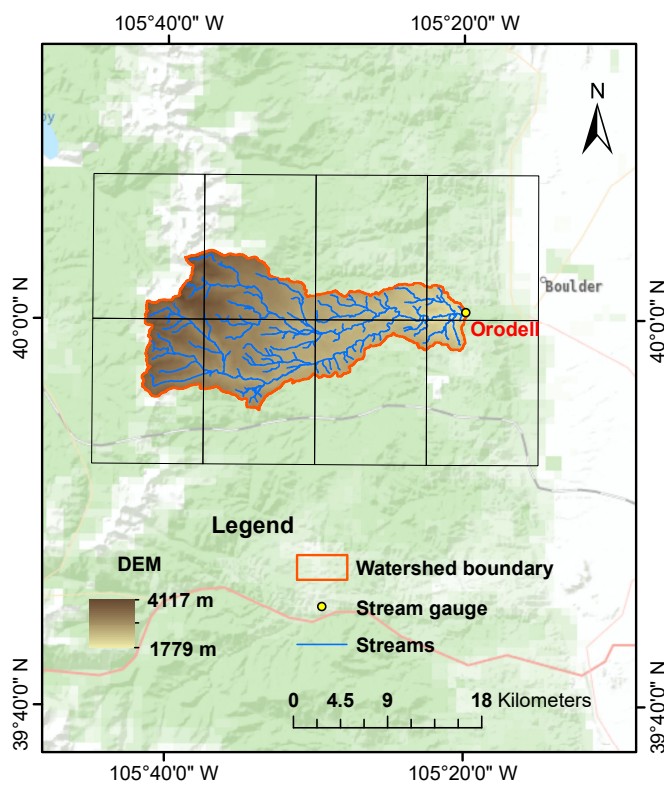

17    **Figure    1.    Geographical    location    of    the    study    area    and    the    1/8°    grid    cells**





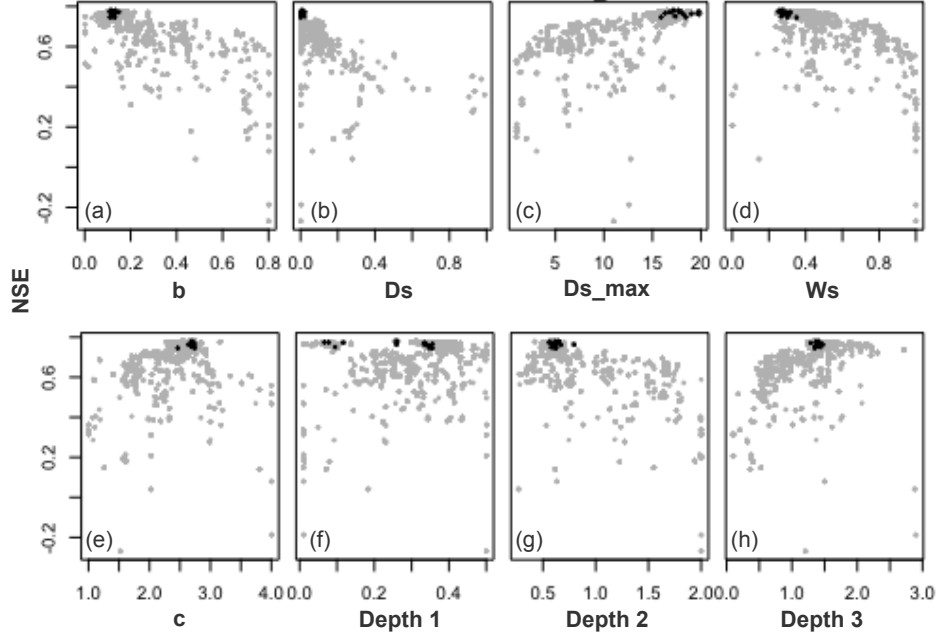

**Figure 2. Parameter sets versus NSE at Boulder Creek. The x-axis represents parameters b, Ds, Ds_max, Ws, c, soil depth 1, soil depth 2, and soil depth 3 in each panel, and was described in Table 1. Grey dots represent all calibration runs (10 000 iterations), black dots represent the 16 optimal parameter sets.**





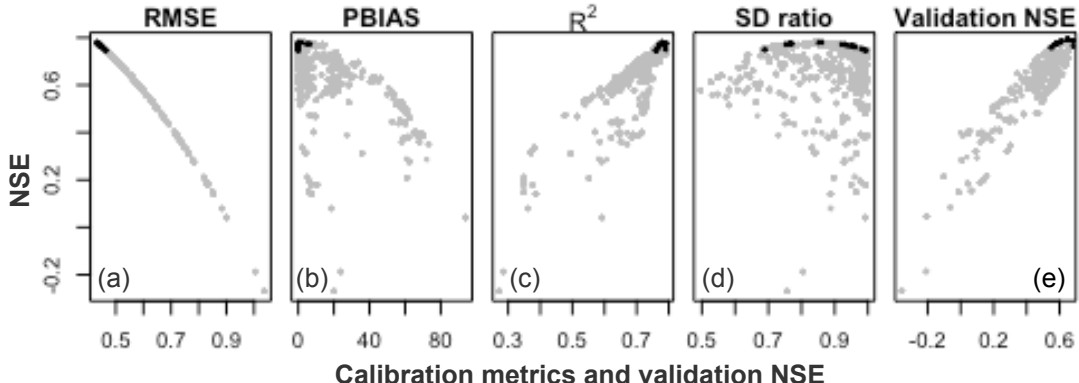

**Calibration metrics and validation NSE**

23    **Figure 3. Calibration metrics and validation NSE at 1/8° spatial resolution at Boulder Creek. The grey dots**
24    **represent all calibration runs, black dots represent the optimal parameter sets.**




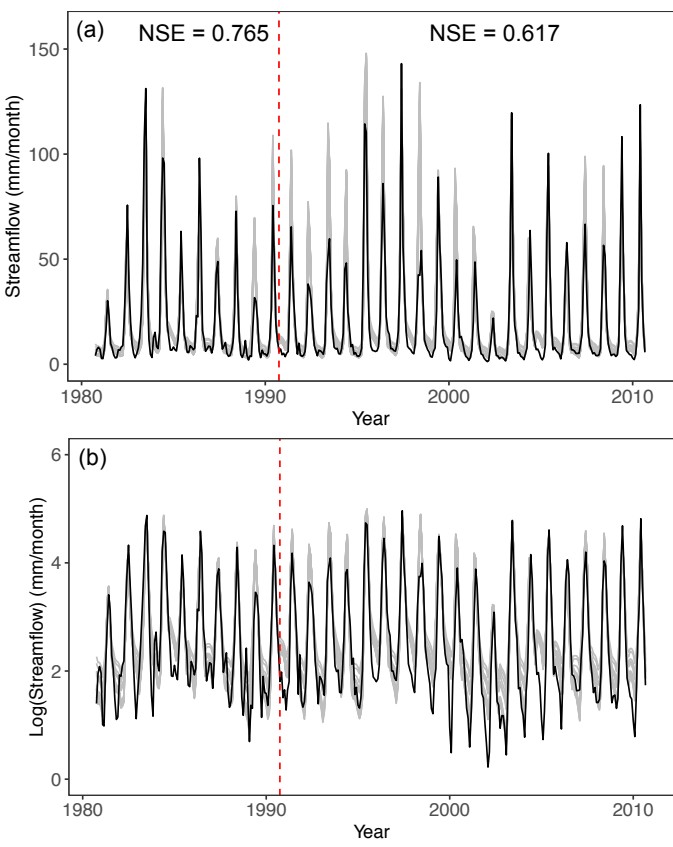

**Figure 4. (a) Monthly simulated (black lines) and observed (grey lines) streamflow at the Boulder Creek**
**Orodell station between 1981 and 2010. (b) Same data shown on a log-scale. The red dashed line separates the**
**calibration and validation periods.**




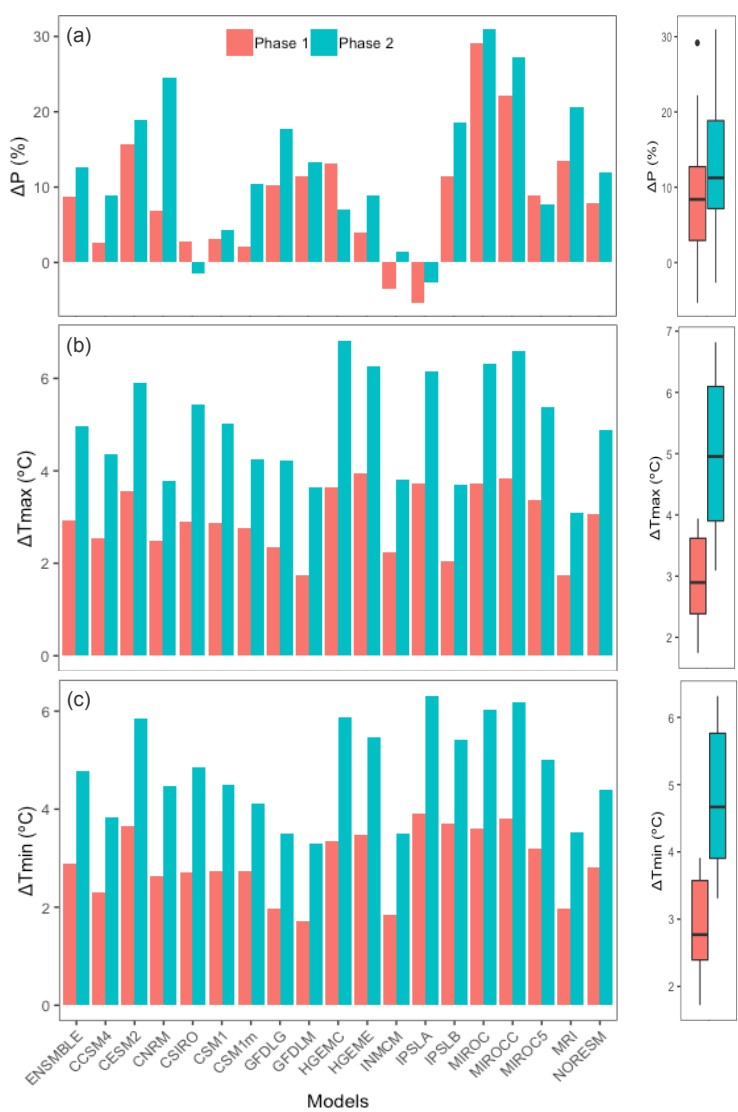

**Figure 5. Changes in average annual precipitation (a), daily maximum air temperature (b) and daily minimum air temperature (c) in Phase 1 (2040-2069) and Phase 2 (2070-2099) compared to the historical period at Boulder Creek. Colored bars correspond to individual climate models and boxplots summarize the variability.**





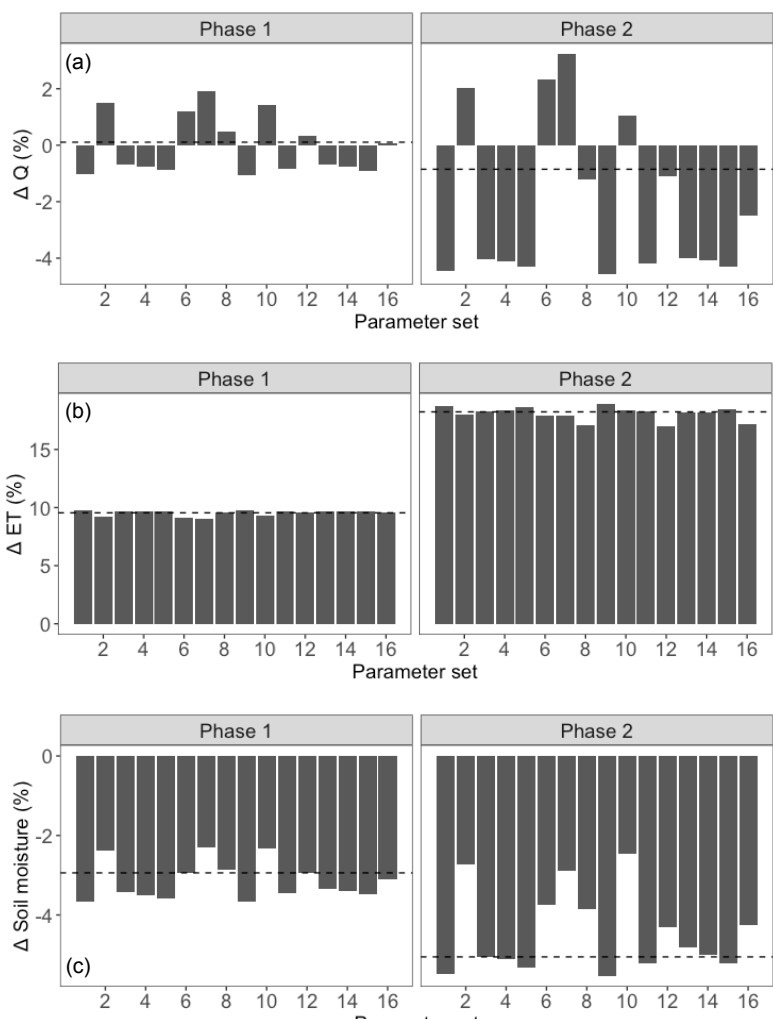

**Figure 6. Median value of changes in average annual streamflow (a), ET (b), and soil moisture (c) using each parameter set in Phase 1 (left column) and Phase 2 (right column). The dashed lines represent median values in annual changes using all parameter sets.**





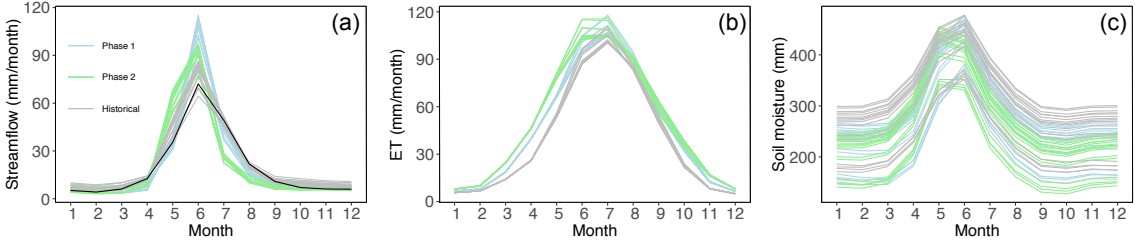

38

**Figure 7. Average monthly values in streamflow (a), ET (b), and soil moisture (c) during the historical time
period, Phase 1, and Phase 2 using the 16 parameter sets. The black line in (a) represents observed data.**





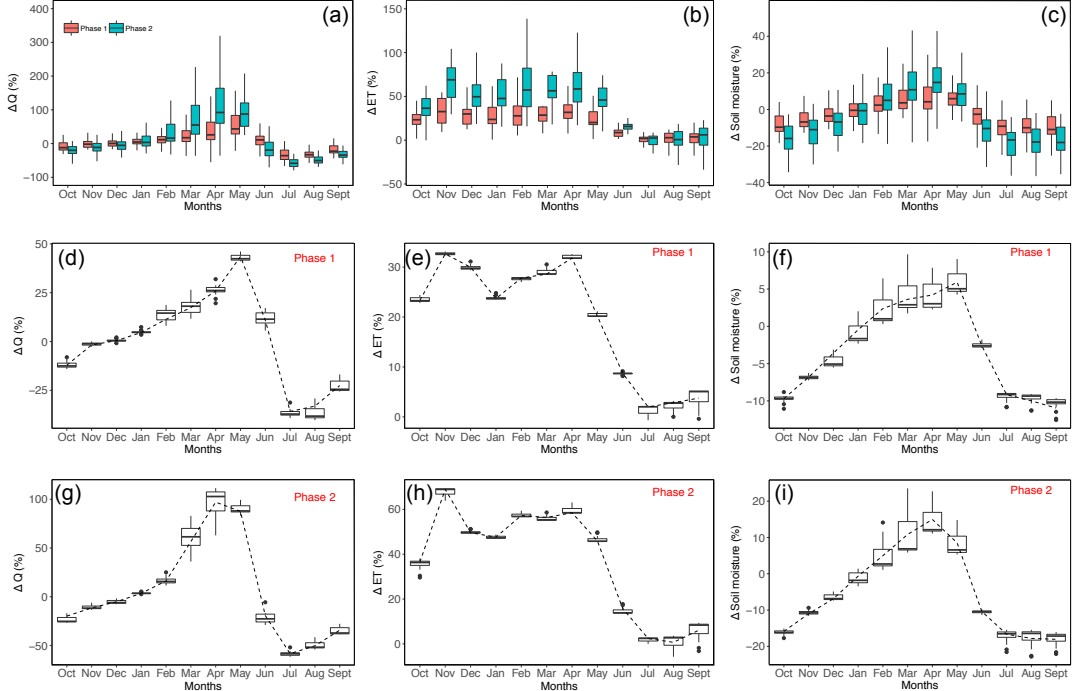

**Figure 8. Boxplots of percent changes using all climate data and parameter sets (a-c), and boxplots of median percent changes using all parameter sets in Phase 1 (d-f) and Phase 2 (g-i) for basin-averaged monthly streamflow, ET, and soil moisture. The dashed lines in the middle and bottom rows represent median changes from all climate data and parameter sets.**