# Peer review of "Contribution of model parameter uncertainty to future hydrological projections"

_Hydrology and Earth System Sciences, 2019_

## Referee Comment (RC1) · Anonymous Referee #1 · 25 Feb 2019

Thank you for the opportunity to review this manuscript. This study deals with an important issue on uncertainty in hydrological models. The authors calibrated the VIC model using a multiobjective approach based on historical data. Sixteen sets of model parameters were randomly selected. Then the model with these 16 sets of parameters were applied to two future phases up to year 2100 with 18 different scenarios. The changes in system states were represented using ensemble means. It is an interesting study. However, it is not very clear what is the major contribution of this study considering the large number of studies on uncertainty in the literature. In addition, some major modeling steps/decisions are not described clearly or justified. Therefore, this reviewer recommends major revision.

Detailed comments: 1. Uncertainty is important as pointed out by the authors. How-

ever, what is the major contribution of this paper considering there are a large number of studies dealing with uncertainty in hydrological models? One of the objectives of the study is to "quantify the uncertainty resultant from model parameters to projections of hydrologic flux and state variables". Has this never been done before? If so, please support with evidence in the introduction section.

2. The context in which parameter uncertainty is assessed. Parameter uncertainty is only one contributor of uncertainties, among model uncertainty, input uncertainty, climate uncertainty, etc. Where does parameter uncertainty sit among all uncertainties?

3. The authors emphasized uncertainty related to Climate Change in introduction. But ensemble mean across different future scenarios are used in the study, where uncertainty represented by the different future scenarios is lost.

4. The authors selected a large number of performance measures. Why were they selected? If they are randomly selected with no justification, the authors are increasing calibration effort without additional benefit.

5. The authors stated that "The 16 best performing parameter sets were chosen randomly using the Borg MOEA framework." This statement is confusing. If the "best" parameter sets are selected, they much be selected based on some criteria rather than randomly. Did the authors mean they are selected randomly from the Pareto-optimal front obtained from multiobjective optimization using Borg? Then why are they selected randomly? Will another different set of parameters selected randomly lead to different results and conclusions?

6. One of the conclusions in the manuscript is that future changes in system state are similar to those in the past (Section 5.3). However, the model was calibrated using historical data then applied to future scenarios. What is the implication of this approach to the conclusion?

7. The authors claimed that "variability due to parameter uncertainty was up to 10 %

annually and 26 % monthly under future climate change scenarios". Without under-standing the whole picture of uncertainty, it will be difficult to reach this conclusion.

Minor comments: 1. Description on the Borg MOEA between lines 148 and 152 is inaccurate.

2. Many terms are used in an ad hoc manner. For example, Borg is an optimization algorithm not a framework. Line 224: "Borg MOEA framework".

3. Line 351 and Line 353: Remove "the" before "summer".

---

## Referee Comment (RC2) · Anonymous Referee #2 · 20 Mar 2019

This paper explored how model parameter uncertainties propagate to future projections. This is an important topic and the simulation experiments are generally well designed. However, I fully concur with the first reviewer that the major contribution of this study is not well identified, given the large number of uncertainty studies already in the literature. Also, some of the key procedures and their rationale should be explained more clearly. Therefore, substantial changes are necessary before the manuscript can be considered for publication.

Major comments:

1. It would be beneficial if the authors would highlight their additional contribution compared to Mendoza et al. (2016), who also concluded that parameter uncertainty could affect the direction and magnitude of projected changes, based on four hydrologic models (including VIC) and three US catchments. Seiller et al. (2017) also examined the effect of parameter uncertainty on future projection, based on multiple catchments, GCM and lumped hydrological models. Could the uncertainty from parameters be compared to uncertainty introduced by different GCMs? What are the minimum parameter sets the authors would recommend for decision-making purpose, considering the computation requirement? By focusing on one model and one catchment, the authors could also carry out more in-depth analyses such as on the plausibility/sensitivity of some parameter values.

2. Based on the results, it is also not quite clear if the authors' general conclusion is fully supported ("multiple optimal parameter sets are needed in order to make meaningful projections of water resource availability into the future"). What projections would be meaningful for this basin, for example, would a few percent increase or decrease in annual flow play a significant role in water supply here, or would the timing be more important? Additionally, the importance of parameter values would be better understood in the context of other uncertainty sources, and previous study showed the relative importance of model structure and parameter values was catchment dependent (Kay et al., 2009). Therefore, it might be worthwhile for the authors to rethink the conclusions from their results.

3. I also have some questions regarding the methodology. In P4L19 "eight parameters were calibrated at 1/8 degree spatial resolution": are there different values on each 1/8 degree grid, or is each parameter the same for the whole basin? Another question is on the climate forcing data. What is the bias correction method used in USBR, and why do the data need to be corrected again by the delta-change method? Would the multiplicative method the authors use lead to some unrealistically high daily precipitation values in future, if the RatioPm is large?

As substantial changes are needed, I would only raise two minor comments at this stage:

- There are numerus grammatical errors in the manuscript such that a careful proof-reading is mandatary. For example in the abstract alone: L16 "parameter sets to"->"parameter sets for"; L23 "result to"->"lead to". - Figure 4. Which period is validation period, which is calibration?

References Kay, A.L., Davies, H.N., Bell, V.A., Jones, R.G. (2009) Comparison of uncertainty sources for climate change impacts: flood frequency in England. Clim. Change 92, 41–63. doi:10.1007/s10584-008-9471-4.

Seiller, G., Roy, R. & Anctil, F. (2017) Influence of three common calibration metrics on the diagnosis of climate change impacts on water resources. J. Hydrol. 547, 280–295. doi:10.1016/j.jhydrol.2017.02.004

---

## Author Comment (AC1) · 14 May 2019

Thank you for the opportunity to review this manuscript. This study deals with an important issue on uncertainty in hydrological models. The authors calibrated the VIC model using a multiobjective approach based on historical data. Sixteen sets of model parameters were randomly selected. Then the model with these 16 sets of parameters were applied to two future phases up to year 2100 with 18 different scenarios. The changes in system states were represented using ensemble means. It is an interesting study. However, it is not very clear what is the major contribution of this study considering the large number of studies on uncertainty in the literature. In addition, some major modeling steps/decisions are not described clearly or justified. Therefore, this reviewer recommends major revision.

[Figure]

Reply: We thank Reviewer 1 for the constructive comments and suggestions that will improve the quality of this work. We have carefully considered all suggestions and outlined a set of proposed revisions in the following response.

Detailed comments: 1. Uncertainty is important as pointed out by the authors. However, what is the major contribution of this paper considering there are a large number of studies dealing with uncertainty in hydrological models? One of the objectives of the study is to "quantify the uncertainty resultant from model parameters to projections of hydrologic flux and state variables". Has this never been done before? If so, please support with evidence in the introduction section.

Reply: We agree that there are a large number of studies dealing with uncertainty in hydrologic models. That said, there are considerably fewer studies that focus specifically on the role of parametric uncertainty in the context of future climate change projections, which is the focus of this paper. We make sure to more clearly articulate this distinction in the revised manuscript. In this context, Dobler et al. (2012) quantified uncertainties resulting from global or regional climate models, bias-correction method, and hydrological model parameterizations in an Alpine watershed of Austria. They concluded that hydrological model parameterization was the least important source of uncertainty. Other studies that have compared multiple uncertainty sources (Addor et al. 2014; Yuan et al. 2017; Joseph et al. 2018) have largely downplayed the importance of model parameterizations. Another study in India went as far as concluding that hydrologic model parametric uncertainty is negligible relative to meteorological uncertainty, (Joseph et al. 2018). If that was indeed true, then it would be justifiable to use any uncalibrated hydrological model to project future hydrological fluxes and state variables. However, a recent study demonstrated that while uncertainty of GCM projections was the dominant source for faster components of hydrologic response like surface runoff, the uncertainty of hydrological model parameterization was found to be a significant source of uncertainty, particularly for slow response components (Her et al., 2019). This finding indicates that calibration of hydrological models is still important

for projecting some important hydrological variables. Another study also demonstrated that model parameters could be of major importance for projecting changes in water-quality (Steffens et al., 2014).

Most relevant to this manuscript, a few studies have investigated how hydrologic modeling decisions can affect future hydrological projections (e.g. Mendoza et al., 2016; Seiller et al., 2017). These studies considered the impacts of the selection of hydrological model, model structure, and parameter sets over a number of river basins. However, what remains unclear, and the focus of this manuscript, is whether the impact of parameter set selection alone is large enough to impact the direction and magnitude of projected changes, and if so, how the magnitude of this impact would translate across different temporal scales. For example, we evaluate whether two model parameter sets that have both been calibrated can produce a different sign in climate sensitivity, e.g. increasing- versus decreasing projected future streamflow. As a result, we focused this work on the contribution of model parameter uncertainty to future hydrological projections, including the contribution of parametric uncertainty at different time scales (annual, monthly, and daily) and for different hydrological variables. Relative to previous studies, we focus here on only a single river basin, but investigate more thoroughly the details of model parameter contributed uncertainty at fine time scales (i.e. daily) and the resultant affects on hydroclimatic extremes.

2. The context in which parameter uncertainty is assessed. Parameter uncertainty is only one contributor of uncertainties, among model uncertainty, input uncertainty, climate uncertainty, etc. Where does parameter uncertainty sit among all uncertainties?

Reply: We agree that parameter uncertainty is only one contributor of uncertainties (see references to Mendoza et al., 2016; Seiller et al., 2017, among others) and that its magnitude relative to other uncertainties can vary. In larger (global) domains, parameter uncertainty might have a smaller impact on future projections due to the potential for compensating errors to partially offset at large spatial scales (Elsner et al., 2014). However, at smaller scales with less potential for errors to cancel out, like the one

presented in this manuscript, the impact of parameter uncertainty has the potential to be large and potentially dominant. In order to evaluate where parameter uncertainty sits among other uncertainties, we compare it with the uncertainty from future climate change scenarios and emphasize this distinction in the revision.

3. The authors emphasized uncertainty related to Climate Change in introduction. But ensemble mean across different future scenarios are used in the study, where uncertainty represented by the different future scenarios is lost.

Reply: We clarify this issue in the revision. Namely that we consider both the ensemble mean—which is widely used for future projections—as well as the ensemble spread as a way to quantify the uncertainty across future climate scenarios. In this way, we aim to preserve the uncertainty represented by different future scenarios. We consider 18 future scenarios and 16 parameter sets to make a total ensemble of 18*16=288 members. The spread of the 288 members is used as a measure of the total uncertainty. For each parameter set, there are 18 future scenarios and the spread of these 18 future scenarios is used to quantify the uncertainty of future scenarios associated with the parameter set. We will therefore obtain the median of the 18 future scenarios for each parameter set and finally get the range from the 16 median values (as for the 16 parameter sets). Following the same approach, we can quantify the uncertainty associated with the different parameter sets. We have clarified the calculation in the revision.

4. The authors selected a large number of performance measures. Why were they selected? If they are randomly selected with no justification, the authors are increasing calibration effort without additional benefit.

Reply: We selected the parameter sets with similar performance but located in different regions of the parameter space following the approaches used by Moriasi et al. (2007) and Demaria et al. (2007). Performance measures were selected for the following specific reasons: NSE is the most commonly used metric in hydrologic modeling and

quantifies the overall performance, putting an emphasis on the seasonal cycle. In contrast, RMSE measures the error in the squared units of the simulated versus observed datasets and emphasizes high flows and outliers. PBIAS measures the average of the simulations compared to the observed datasets, evaluating the quality of the overall water balance. R2 is the coefficient of determination between observed and simulated datasets. The ratio of standard deviations exclusively evaluates the variability of the model relative to the observation and thus represents a valuable quality control. We now clarify this rationale further in the revision.

5. The authors stated that "The 16 best performing parameter sets were chosen randomly using the Borg MOEA framework." This statement is confusing. If the "best" parameter sets are selected, they much be selected based on some criteria rather than randomly. Did the authors mean they are selected randomly from the Pareto-optimal front obtained from multiobjective optimization using Borg? Then why are they selected randomly? Will another different set of parameters selected randomly lead to different results and conclusions?

Reply: We thank the reviewer for making this point. Yes, we chose the parameters from the Pareto optimal front, so as to represent non-dominated solutions. That is, parameter sets with similar performance but located in different regions of the parameter space were chosen. The Borg MOEA uses the epsilon non-dominance operator, which has the advantages of convergence and diversity with respect to approximating the true Pareto-optimal front over other MOEA. The epsilons represent the resolutions of the objective functions. Specifically, the epsilon-box dominance archive divides the objective space into hyper-boxes with side-length epsilon, so called epsilon-boxes (Hadka and Reed, 2013). The 16 parameter sets represent epsilon-box non-dominated solutions that were sampled from the full Pareto-optimal front such that the conclusions here are expected to be robust. The selecting principle and method have been clarified in the revision.

6. One of the conclusions in the manuscript is that future changes in system state

are similar to those in the past (Section 5.3). However, the model was calibrated using historical data then applied to future scenarios. What is the implication of this approach to the conclusion?

Reply: I think you refer to L354-357. The implicit assumption here is that parameter sets calibrated during historical periods can be applied to future simulations. The chosen parameter sets give satisfactory results during validation period historically. The focus here is how uncertainties across 'calibrated' parameter sets will change between past conditions relative to future climate. We make this assumption clearer in the revised manuscript.

7. The authors claimed that "variability due to parameter uncertainty was up to 10 % annually and 26 % monthly under future climate change scenarios". Without understanding the whole picture of uncertainty, it will be difficult to reach this conclusion.

Reply: Please see our response to #3 above and associated clarifications in the manuscript. These numbers represent the fraction (or percentage) of the uncertainty associated with parametric spread relative to the total (parametric + climate scenario) uncertainty.

Minor comments: 1. Description on the Borg MOEA between lines 148 and 152 is inaccurate.

Reply: We have updated the description of the Borg MOEA for accuracy.

2. Many terms are used in an ad hoc manner. For example, Borg is an optimization algorithm not a framework. Line 224: "Borg MOEA framework".

Reply: We correct and streamline the terminology and usage in the updated version.

3. Line 351 and Line 353: Remove "the" before "summer".

Reply: Will be corrected.

References

Addor, N., O. Rössler, N. Köplin, M. Huss, R. Weingartner, and J. Seibert (2014), Robust changes and sources of uncertainty in the projected hydrological regimes of Swiss catchments, Water Resour. Res., 50, doi:10.1002/2014WR015549.

Demaria, E. M., Nijssen, B., and Wagener, T. (2007) Monte Carlo sensitivity analysis of land surface parameters using the Variable Infiltration Capacity model. Journal of Geophysical Research, 112, D11113, doi:10.1029/2006JD007534.

Dobler, C., Hagemann, S., Wilby, R. L., and Stötter, J.: Quantifying different sources of uncertainty in hydrological projections in an Alpine watershed, Hydrol. Earth Syst. Sci., 16, 4343-4360, https://doi.org/10.5194/hess-16-4343-2012, 2012.

Elsner, M. M., S. Gangopadhyay, T. Pruitt, L. D. Brekke, N. Mizukami, and M. P. Clark (2014) How Does the Choice of Distributed Meteorological Data Affect Hydrologic Model Calibration and Streamflow Simulations? Journal of Hydrometeorology, 15(4):1384-1403, doi:10.1175/JHM-D-13-083.1.

Hadka, D., Reed, P. (2013) Borg: An Auto-Adaptive Many-Objective Evolutionary Computing Framework. Evolutionary Computation, 21, 231-259, doi: 10.1162/evco_a_00075.

Her, Y., Yoo, S. H., Cho, J., Hwang, S., Jeong, J., & Seong, C. (2019). Uncertainty in hydrological analysis of climate change: multi-parameter vs. multi-GCM ensemble predictions. Scientific reports, 9(1), 4974.

Joseph, J., Ghosh, S., Pathak, A., & Sahai, A. K. (2018) Hydrologic impacts of climate change: Comparisons between hydrological parameter uncertainty and climate model uncertainty. Journal of Hydrology, 566, 1-22.

Mendoza, P. A., M. P. Clark, N. Mizukami, A. J. Newman, M. Barlage, E. D. Gutmann, and R. M. Rasmussen (2015) Effects of Hydrologic Model Choice and Calibration on the Portrayal of Climate Change Impacts. Journal of Hydrometeorology, 16:762-780, doi: 10.1175/JHM-D-14-0104.1.
Moriasi, D. N., Arnold, J. G., Van Liew, M. W., Bingner, R. L., Harmel, R. D., and Veith, T. L. (2007) Model evaluation guidelines for systematic quantification of accuracy in watershed simulations. American Society of Agricultural and Biological Engineers, 50(3):885-900.

Steffens, K., Larsbo, M., Moeys, J., Kjellström, E., Jarvis, N., and Lewan, E.: Modelling pesticide leaching under climate change: parameter vs. climate input uncertainty, Hydrol. Earth Syst. Sci., 18, 479-491, https://doi.org/10.5194/hess-18-479-2014, 2014.

Younggu Her, Seung-Hwan Yoo, Jaepil Cho, Syewoon Hwang, Jaehak Jeong & Chounghyun Seong, Uncertainty in hydrological analysis of climate change: multi-parameter vs. multi-GCM ensemble predictions, Scientific Reports, 9, 4974 (2019).

Yuan, F., C. Zhao, Y. Jiang, L. Ren, H. Shan, L. Zhang, Y. Zhu, T. Chen, S. Jiang, X. Yang, H. Shen: Evaluation on uncertainty sources in projecting hydrological changes over the Xijiang River basin in South China, J. Hydrol., 554 (2017), pp. 434-450.

Please also note the supplement to this comment: https://www.hydrol-earth-syst-sci-discuss.net/hess-2019-52/hess-2019-52-AC1-supplement.pdf

---

## Author Comment (AC2) · 14 May 2019

This paper explored how model parameter uncertainties propagate to future projections. This is an important topic and the simulation experiments are generally well designed. However, I fully concur with the first reviewer that the major contribution of this study is not well identified, given the large number of uncertainty studies already in the literature. Also, some of the key procedures and their rationale should be explained more clearly. Therefore, substantial changes are necessary before the manuscript can be considered for publication.

Reply: We thank Reviewer 2 for the thoughtful review that will improve the quality of this work. We have carefully considered all suggestions and outlined a set of proposed

revisions in the following response.

Major comments:

1. It would be beneficial if the authors would highlight their additional contribution compared to Mendoza et al. (2016), who also concluded that parameter uncertainty could affect the direction and magnitude of projected changes, based on four hydrologic models (including VIC) and three US catchments.

Reply: We thank the reviewer for raising this important distinction and have made additional clarification of the additional contribution of this paper relative to Mendoza et al. (2016) (and others) in the revision. Mendoza et al. (2016) looked at the effects of model structure, objective function, multiple local optima, and forcing calibration dataset to the projections of flux and state variables in several basins at monthly time scale. In this study, we applied downscaled climate data from 18 GCM models and chose two future phases (Phase 1 is 2040-2069 and Phase 2 is 2070-2099). We then quantified the projection uncertainty from both model parameters as well as scenario choice, for the Boulder Creek Watershed. Our study is different from Mendoza et al. (2016) in several aspects: 1) our study used multiple future scenarios from GCMs—enabling a broader characterization of the relative uncertainty from scenario choice—rather than applying a single future pseudo global warming scenario (adding a mean climate perturbation to historical conditions) in the case of Mendoza, thus our study can more fully put parametric uncertainty in the context of future scenario uncertainty, 2) our study evaluates the effect of uncertainty at different time scales (annual, monthly, and daily) rather than the exclusively monthly time scale analysis of Mendoza, thus enabling our study to provide insights to the impacts of the respective uncertainty sources at shorter time scales and thus may shed light on the impacts on hydroclimatic extremes that generally occur at shorter-than-monthly timescales.

Seiller et al. (2017) also examined the effect of parameter uncertainty on future projection, based on multiple catchments, GCM and lumped hydrological models. Could

the uncertainty from parameters be compared to uncertainty introduced by different GCMs?

Reply: Yes, we thank the reviewer for this suggestion, this is correct, we can directly compare the magnitude of parametric uncertainty to the uncertainty that results from considering different GCMs. Most studies to date, have ignored the role of parametric uncertainty on climate change sensitivity. Among the comparatively few studies that do consider parametric uncertainty in future projection, the majority demonstrate that the uncertainty introduced by different GCMs was higher than that from parameters. In the revision, we now directly compare the uncertainty from parameters to uncertainty introduced by different GCMs. Seiller et al. (2017) investigated the effects of calibration metrics (i.e. selecting different objective functions) on future projection. However, our study would argue that even calibration with the same objective functions that produce largely similar historical performance, large differences can be seen in hydrologic response to future projections.

What are the minimum parameter sets the authors would recommend for decision-making purpose, considering the computation requirement?

Reply: We have added a brief discussion on this topic in the revision. This study seeks to foremost advance understanding into the contribution of parameter uncertainty to future hydrologic projections, with the most profound result being that in this watershed, different parameter sets can produce different directions in hydrologic changes. For a situation where the parametric ensemble shows both increasing and decreasing response, we might recommend to a decision-maker that the change sensitivity is not robust, since the ensemble includes positive and negative changes, although the mean or median change may be positive or negative. For this analysis, we have selected a small river basin so as to avoid the situation where compensating errors may offset each other, e.g. where increasing and decreasing responses may cancel each other out, as well as to ensure that computation capacity for calibration was not a limitation for the experiments. We did not examine whether an increase in the number of param-
eter sets would reduce 'uncertainty' or not. Even if the ensemble mean values would become stable when the number of parameter sets became large, it would be difficult to conclude that the ensemble mean value would be the best "decision recommendation" for future sensitivities. This challenge is within the realm of a more fundamental question: whether we can treat all parameter sets equally or not? While we agree (and articulate in the revision) that proposing a minimum number of parameter sets would be useful for decision makers, we also posit that this may be catchment dependent and more importantly it would depend on the timescale of interest (e.g. flooding versus drought), which is a larger question worthy of a separate study.

By focusing on one model and one catchment, the authors could also carry out more in-depth analyses such as on the plausibility/sensitivity of some parameter values.

Reply: We would like to thank the reviewer for this suggestion. The sensitivity of some parameter values may be very informative for understanding the role of each parameter. In the revision we have added a supplemental analysis into the role of individual parametric sensitivity to the overall model sensitivity, using a variance-based approach. While the focus of the manuscript remains on the parameter sensitivity to future climate change, the parametric sensitivity helps us to understand whether this change can be most easily attributed to a single parameter or whether it's more equally distributed across those parameters that were calibrated.

2. Based on the results, it is also not quite clear if the authors' general conclusion is fully supported ("multiple optimal parameter sets are needed in order to make meaningful projections of water resource availability into the future"). What projections would be meaningful for this basin, for example, would a few percent increase or decrease in annual flow play a significant role in water supply here, or would the timing be more important?

Reply: We would like to thank the reviewer for this suggestion. We agree that the general conclusion is not well written (or overstated), and we have modified it in the

revision. Our primary result here is that different parameter sets can result in large difference in future projections. The results demonstrate the need to consider multiple optimal parameter sets in order to provide a more complete range of future projections of water resource availability into the future. Boulder Creek is an important water source as it provides drinking water, agricultural irrigation, aquatic habitat, recreation, and so forth. Both the timing and magnitude of flows matter in this watershed (e.g. Murphy et al., 2003). Accordingly, we revise the manuscript to more thoroughly contextualize how the changes shown by this work would affect water resources and decisions here and in similar basins.

Additionally, the importance of parameter values would be better understood in the context of other uncertainty sources, and previous study showed the relative importance of model structure and parameter values was catchment dependent (Kay et al., 2009). Therefore, it might be worthwhile for the authors to rethink the conclusions from their results.

Reply: We agree that the importance of parameter values would be better understood in the context of other uncertainty sources. We acknowledge that uncertainties due to model structure, objective functions, and hydrological indicators can be catchment dependent. Here, we chose a topographically complex watershed to investigate the contribution of model parameter uncertainty relative to GCM uncertainty. Both the type of watershed, as well as the (only) two sources of uncertainty examined have implications for our conclusions, e.g. are most relevant for snowmelt dominated montane domains. Furthermore, we evaluate the role of uncertainty at different time scales (annual, monthly, and daily) and for different hydrological variables. We now interpret our results within the context of Kay et al. (2009) and have revised the manuscript to avoid overstatements in conclusions.

3. I also have some questions regarding the methodology. In P4L19 "eight parameters were calibrated at 1/8 degree spatial resolution": are there different values on each 1/8 degree grid, or is each parameter the same for the whole basin? Another question is

on the climate forcing data. What is the bias correction method used in USBR, and why do the data need to be corrected again by the delta-change method? Would the multiplicative method the authors use lead to some unrealistically high daily precipitation values in future, if the RatioPm is large?

Reply: We thank the reviewer for raising these issues. To clarify, each parameter set is applied over the entire basin. The bias-correction and spatial disaggregation (BCSD) climate data from USBR used a quantile mapping method on location-specific datasets, between gridded observations (Maurer et al., 2002) and the GCM historical data. For value in each grid cell and variable, cumulative distribution functions (CDFs) of conditions from both observed and GCM historical datasets were constructed. At each percentile rank in the GCM historical CDFs, observed values in the same rank were identified and applied so that the historical GCM and observed CDFs match. Quantile mapping was performed (by USBR) by populating the sample distribution using a 15-day moving window centered on each calendar day. Quantile mapping adjustments from historical GCM and observed datasets are then transferred to future time slices (USBR, 2013; Abatzoglou and Brown, 2012). The RatioPm varies from 0.76 to 1.39, with an average value of 1.02. The values have seasonal fluctuations, but generally center around 1.02. Using this adjusted dataset from USBR, we calculated and applied the delta change method, e.g. the difference in future climate relative to historical, in order to impose regional climate variability on the model. We have clarified the methodology in the revised manuscript.

As substantial changes are needed, I would only raise two minor comments at this stage:

- There are numerus grammatical errors in the manuscript such that a careful proof-reading is mandatary. For example in the abstract alone: L16 "parameter sets to"->"parameter sets for"; L23 "result to"->"lead to". - Figure 4. Which period is validation period, which is calibration?

Reply: Thank you for your suggestions, we will thoroughly proofread the revised manuscript. In Figure 4, the validation period is 1991-2010 and the calibration period is 1981-1990, which are specified in L12-14 on Page 5. The different lengths of the calibration and validation time periods refer to the similar approach in Bastola et al. (2011). We will include this information in the updated figure caption.

References Kay, A.L., Davies, H.N., Bell, V.A., Jones, R.G. (2009) Comparison of uncertainty sources for climate change impacts: flood frequency in England. Clim. Change 92, 41–63. doi:10.1007/s10584-008-9471-4.

Seiller, G., Roy, R. & Anctil, F. (2017) Influence of three common calibration metrics on the diagnosis of climate change impacts on water resources. J. Hydrol. 547, 280–295. doi:10.1016/j.jhydrol.2017.02.004

Additional references:

Abatzoglou, J. T., and Brown, T. J. (2012) A comparison of statistically downscaling methods suited for wildfire applications. International Journal of Climatology, 32:772-780, doi: 10.1002/joc.2312.

Bastola, S., Murphy, C., and Sweeney, J. (2011) The role of hydrological modelling uncertainties in climate change impact assessments of Irish river catchments. Advances in Water Resources, 34:562-576, doi:10.1016/j.advwatres.2011.01.008.

Maurer, E. P., Wood, A. W., Adam, J. C., Lettenmaier, D. P., and Nijssen, B. (2002) A Long-Term Hydrologically Based Dataset of Land Surface Fluxes and States for the Conterminous United States. Journal of Climate, 15:3237-3251, https://doi.org/10.1175/1520-0442(2002)015<3237:ALTHBD>2.0.CO;2.

Murphy, S. F., Barber, L. B., Verplanck, P. L., and Kinner, D. A. (2003) Environmental setting and hydrology of the Boulder Creek Watershed, Colorado. In S.F. Murphy et al. (ed.) Comprehensive water quality of the Boulder Creek Watershed, Colorado during high-flow and low-flow conditions, 2000: U.S. Geological Survey Water-Resources

Investigations Report 03- 4045, pp. 5-26.

USBR (2013) Downscaled CMIP3 and CMIP5 Climate and Hydrology Projections: Release of Downscaled CMIP5 Climate Projections, Comparison with Preceding Information, and Summary of User Needs. U.S. Department of the Interior, Bureau of Reclamation, Technical Services Center, Denver, Colorado, 47pp.

Please also note the supplement to this comment:
https://www.hydrol-earth-syst-sci-discuss.net/hess-2019-52/hess-2019-52-AC2-supplement.pdf